# Phytoextraction Potential of Sunn Hemp, Sunflower, and Marigold for Carbaryl Contamination: Hydroponic Experiment

**DOI:** 10.3390/ijerph192416482

**Published:** 2022-12-08

**Authors:** Najjapak Sooksawat, Duangrat Inthorn, Apisit Chittawanij, Alisa Vangnai, Pornpimol Kongtip, Susan Woskie

**Affiliations:** 1Department of Agricultural Engineering and Technology, Faculty of Agriculture and Natural Resources, Rajamangala University of Technology Tawan-Ok, Chonburi 20110, Thailand; 2Center of Excellence in Agricultural Machinery, Faculty of Agriculture and Natural Resources, Rajamangala University of Technology Tawan-Ok, Chonburi 20110, Thailand; 3Department of Environmental Health Sciences, Faculty of Public Health, Mahidol University, Bangkok 10400, Thailand; 4Center of Excellence on Environmental Health and Toxicity (EHT), Bangkok 10400, Thailand; 5Department of Plant Production, Faculty of Agriculture and Natural Resources, Rajamangala University of Technology Tawan-Ok, Chonburi 20110, Thailand; 6Department of Biochemistry, Faculty of Sciences, Chulalongkorn University, Bangkok 10330, Thailand; 7Center of Excellence in Biocatalyst and Sustainable Biotechnology, Chulalongkorn University, Bangkok 10330, Thailand; 8Department of Occupational Health and Safety, Faculty of Public Health, Mahidol University, Bangkok 10400, Thailand; 9Department of Public Health, University of Massachusetts Lowell, Lowell, MA 01854, USA

**Keywords:** flower plants, toxicity, bioaccumulation, carbaryl

## Abstract

The phytoextraction ability and responses of sunn hemp, sunflower, and marigold plants were investigated toward carbaryl insecticide at 10 mg L^−1^ and its degradative product (1-naphthol). All test plants exhibited significant carbaryl removal capability (65–93%) with different mechanisms. Marigold had the highest translocation factor, with carbaryl taken up, translocated and accumulated in the shoots, where it was biotransformed into 1-naphthol. Consequently, marigold had the least observable toxicity symptoms caused by carbaryl and the highest bioconcentration factor (1848), indicating its hyperaccumulating capability. Sunflower responded to carbaryl exposure differently, with the highest carbaryl accumulation (8.7 mg kg^−1^) in roots within 4 days of cultivation, leading to a partial toxicity effect. Sunn hemp exhibited severe toxicity, having the highest carbaryl accumulation (91.7 mg kg^−1^) that was biotransformed to 1-naphthol in the sunn hemp shoots. In addition, the different models were discussed on plant hormone formation in response to carbaryl exposure.

## 1. Introduction

Pesticides are widely used for pest control on farmland for crop and livestock protection and even in residential areas for the health of humans and their pets. Carbaryl (chemical name 1-naphthalenyl methyl carbamate) is a carbamate pesticide that is a widely used broad-spectrum insecticide. Carbaryl is used in agriculture, horticulture, and residential settings on corn, soybean, cotton, fruit, nut, vegetable crops, and rice, as well as in-home yards and gardens [1]. Carbaryl is used on intensive rice farmland for the control of grasshoppers and crickets that damage grasses and other vegetation by consuming the stems and leaves. Insect feeding causes direct damage to rice plant growth and seed production, thus reducing valuable grain productivity [2]. After the rice is transplanted at around 30 days, it has green leaves, which are a feed source for grasshoppers; subsequently, the rice blooms at around 90 days, with the flower inflorescence also being attractive to grasshoppers and other insects. Thus, following rice transplantation, a carbaryl application is sprayed at around 30 or 90 days or both, depending on the presence of insects, such as grasshoppers. Some carbamate pesticides, such as carbosulfan, may be applied as well if insects, such as aphids, are present [3]. In 2014, a survey of carbaryl use in agriculture in Mae Taeng district, Chiang Mai, Thailand, was conducted by the Environmental Research and Training Center that reported carbaryl was used by farmers for pest control in the area, including as a herbicide, fertilizer, and plant hormone [4]. Water sampling from the area showed carbamate residue present at 0.01 mg L^−1^. Soil sampling detected carbaryl at <0.02 mg L^−1^, and plant sampling recorded carbamate residue (methomyl) during the harvesting period of the yardlong bean as high as 0.02–0.08 mg L^−1^ and in soybean sprout as high as 0.53 mg L^−1^ [4].

Carbaryl has a water solubility of 36 mg L^−1^ at 20 °C [2]. It moves and is transformed in the environment. The environmental fate processes include persistence and degradation, mobility and migration potential to groundwater and surface water, and plant uptake. Its degradation in aerobic soil varies from rapid to slow, with a half-life ranging from 4 to 253 days, depending on the pH and aerated soil conditions. 1-naphthol is the major degradation of carbaryl under aerobic and anaerobic conditions in soil and water. Sorption of 1-naphthol to soil also increases with increasing organic carbon content. The degradation of 1-naphthol is expected to be less persistent in the field than carbaryl [2].

Acute (short-term) and chronic (long-term) exposure of humans to carbaryl has been observed to cause cholinesterase inhibition. The reduced levels of this enzyme in the blood cause neurological effects. These effects appear to be reversible upon discontinuation of exposure. Headache, memory loss, muscle weakness and cramps, and anorexia result from cholinesterase inhibition caused by prolonged low-level exposure to carbaryl [1]. In 2008, the WHO announced that the acceptable daily intake of carbaryl is 0–0.008 mg kg^−1^ of body weight [5]. Ecological toxicity in terrestrial and aquatic plants and animals has been observed and reported for carbaryl [2]. In 2013, the Ministry of Agriculture and Cooperatives, Thailand, announced a standard for maximum residue limits (MRL) for carbaryl in agricultural products as <0.02–10 mg kg^−1^ [6]. The MRL for rice is 1 mg kg^−1^ [6]. Unfortunately, carbaryl has been found at unsafe levels in some vegetables [7]. In 2019, USDA reported that carbaryl could react with nitrile under certain conditions to produce N-nitrosocarbaryl, which has been shown to be carcinogenic and mutagenic in laboratory test systems [2].

Carbaryl degrades readily to 1-naphthol, which is several times more toxic than its precursor. Despite its lower persistence in the environment and its wide distribution, there is a need to know whether and to what extent carbaryl is present because of its potentially toxic effects [8]. When present in the environment at unusually high levels, biodegradable chemicals can be extremely persistent, which increases the risk of surface and groundwater contamination. Thus, contaminated soil at agrochemical facilities requires quick, efficient and economical cleanup. Phytoremediation is one method that can remediate pesticide residue in the soil. Plants remediate organic contaminants via three mechanisms: (1) the direct uptake of contaminants and the subsequent accumulation of nonphytotoxic metabolites into plant tissue; (2) the release of exudates and enzymes that stimulate microbial activity and biochemical transformations; and (3) the enhancement of mineralization in the rhizosphere (the root-soil interface), which is attributable to mycorrhizal fungi and the microbial consortia [3,9]. Phytoremediation has been successfully used to clean up persistent organic pesticides such as atrazine, alachlor and metolachlor [10]. The removal efficiency of several plants has been reported regarding pesticides, such as carbofuran, chlorpyrifos, carbaryl, inuron, permethrin, and triazophos [3,11,12,13]. Some species, such as *Phragmites* sp., can remove pesticides [14,15,16], while plants, such as *Phragmites australis*, have been used to remove pesticide wastewater based on a wetland system [14,17]. Among these remediating plants, some can be classified as ornamental and leguminous, such as lupine, sunflower, and morning glory. The leguminous plant, *Lupinus angustifolius,* is tolerant to the uptake or phytoextraction of carbaryl [9,12], while sunflower can also accelerate the degradation of carbamate pesticides in soil [3]. Although marigold (*Tagetes erecta*) and pot marigold (*Calendula officinalis*) have shown evidence of phytoremediation in heavy metals [18,19], there has been little reported on using marigold to remediate carbamate pesticide. These tropical plants, sunn hemp (*Crotalaria juncea* L.), sunflower (*Helianthus annuus* L.) and marigold (*Tagetes erecta*), have a short life cycle and may be suitable for on-farm crop rotation in crop fields, especially as agro- and eco-tourist attractions. 

Phytoextraction is the process whereby plant tissue helps to remove pollutants by direct uptake. Testing for hyperaccumulators for application in the field is important. Several plant species may be used on-site to remove more than one contaminant [16,20,21]. Plants that have the ability to perform phytoextraction are characterized by their ability to accumulate and tolerate high concentrations of pollutants while maintaining a rapid growth rate. Thus, the present research investigated using phytoextraction of a synthetic solution treated with carbaryl pesticide. The factors examined were: toxicity (in terms of relative growth rate, plant morphology and pigment contents), bioaccumulation (in terms of bioconcentration factor, translocation factor and enrichment factor), and removal efficiency (in terms of uptake capacity for carbaryl residue under hydroponic conditions).

## 2. Materials and Methods

### 2.1. Plant Materials and Cultivation 

Seeds of sunn hemp (*C. juncea* L.), sunflower (*H. annuus* L.) and marigold (*T. erecta*) were obtained from Chatuchak, Bangkok, Thailand. The plants were grown in the laboratory and examined for their phytoremediation ability. 

In the developed hydroponic system, all plant seeds were germinated in purified water on a soaking sponge for 3–4 days. Then the plants were grown in a hydroponic culture for 2–3 weeks with 30% Hoagland’s s solution. At a starting amount of 23–45 g L^−1^, the plants were transferred to grow in nutrients containing 0, 5, or 10 mg carbaryl in 1000 mL of nutrient solution [12]. The control was a nutrient solution without a plant at pH 6 adjusted with 1 N NaOH and 1 N HCl, a light intensity of approximately 10,000 lux, and daylight:darkness photoperiod of 10 h:14 h at 28 ± 2 °C in a glass container in the laboratory. The control (non-planted nutrient solution) and plant samples were grown in triplicate. After cultivating and harvesting at 0, 4, 8 and 12 d, a 1 g sample of each plant species was collected. Carbaryl and its degradation, 1-naphthol, were determined from the plant and nutrient solution using a high-performance liquid chromatography (HPLC) technique.

### 2.2. Study of Carbaryl Toxicity in Hydroponic System 

Plants were determined for symptoms of toxicity following their exposure to carbaryl (0, 5, or 10 mg L^−1^) for 4, 8, and 12 days. Plant material was moved, rinsed with distilled water, and separated into shoots and roots, and the fresh weight of each sample was determined. The total chlorophyll, chlorophyll a, chlorophyll b, and carotenoid contents were determined for toxicity of the pigment content [22]. Samples (each 1.5 g) of plant shoots cultivated for 4, 8, and 12 days were ground, and pigment extraction was performed using 7.5 mL of 80% acetone for 2 min. Then, 5 mL of the solution was transferred and centrifuged at 3000 rpm for 3 min to separate the pellet. Triplicate samples were taken and determined for chlorophyll a, b and carotenoid at wavelengths of 663, 645, and 480 nm, respectively, using a spectrophotometer [22]. Carbaryl and 1-naphthol were determined separately for the plant material and the nutrient solution. The relative growth rate (RGR), shoot and root lengths shoot and root weights, and leaf number were examined [11]. 

The RGR (mg g^−1^ fresh weight day) was calculated for total biomass based on Equation (1):(1)$$\mathrm{RGR}=\frac{{\mathrm{lnW}}_{2}-{\mathrm{lnW}}_{1}}{t_{2}-t_{1}}\;\times 100$$
where W_1_ (g) and W_2_ (g) are the initial and final plant fresh weights, respectively and *t*_1_ (days) and *t*_2_ (days) are the initial and final times.

### 2.3. Study of Carbaryl Accumulation and Removal

Each plant was rinsed, cut, and group selected into shoots and roots. Carbaryl and 1-naphthol were determined in both plant samples and nutrient solutions. Shoot and root accumulation of carbaryl, uptake capacity, bioconcentration factor (BCF), translocation factor (TF), and enrichment factor (EF) for carbaryl residue were examined [18]. 

The uptake capacity of carbaryl in the hydroponic system was calculated for carbaryl removal based on Equation (2):(2)$$\mathrm{Uptake}\;\mathrm{capacity}=\frac{C_{1}-C_{2}}{M}\;\times \;V$$
where *C*_1_ is the initial concentration of carbaryl in the aqueous solution, *C*_2_ is the concentration of carbaryl in the aqueous solution at the final time of measurement, V is the volume of the solution, and *M* is the plant biomass.

BCF, TF, EF, and the removal percentage were determined using both shoots and roots of plant material based on Equations (3)–(6):(3)$$\mathrm{BCF}=\frac{Concentration\;in\;plant\;at\;harvest}{Concentration\;in\;nutrient\;solution\;at\;harvest}$$
(4)$$\mathrm{TF}=\frac{Concentration\;in\;shoot\;at\;harvest}{Concentration\;in\;root\;at\;harvest}$$
(5)$$\mathrm{EF}=\frac{Concentration\;in\;shoot\;at\;harvest}{Concentration\;in\;nutrient\;solution}$$
(6)$$\mathrm{Removal}\;(\%)=\frac{Concentration\;in\;nutrient\;at\;t_{1}-Concentration\;in\;nutrient\;at\;t_{2}}{Concentration\;in\;nutrient\;at\;t_{1}}\;\times \;100$$

### 2.4. Determination of Carbaryl and 1-Naphthol 

The carbaryl and 1-naphthol were HPLC grade and purchased from Sigma-Aldrich, Burlington, MA, USA and Supelco, Bellefonte, PA, USA, respectively. Carbaryl and 1-naphthol can be effectively determined using HPLC-photo-diode array (PDA) detection with lower levels of carbaryl [8,23]. The preparation of a standard solution of carbaryl, the sample extraction procedure, and instrumental conditions for HPLC analysis has been described by Biswas and colleagues and Ozhan and colleagues [8,23]. The detection limit was below 0.8 μg L^−1^. The standard solution of carbaryl was freshly prepared in at least 5 concentrations (0.02–20.0 mg L^−1^), and the standard solution of 1-naphthol was fresh-prepared in at least 5 concentrations (0.005–2.0 mg L^−1^). The procedure of carbaryl and 1-naphthol determination was modified by Biswas and colleagues and Ozhan and colleagues [8,16]. Plant samples were cut and separated into shoots and roots, weighed, and the weights were recorded. Then, 1 g of each sample was ground and extracted for carbaryl and 1-naphthol with 3 mL of acetonitrile for 15 min at room temperature. The mixture was vortexed for 1.5 min and left for 5 min before adding 2 mL of distilled water and vortexing for 1 min. The mixture was centrifuged at 3000 rpm for 15 min, and the supernatant was collected. The sample was passed through a 0.2-µm filter and kept in the freezer for no longer than 2 weeks before the HPLC analysis. For the determination of carbaryl and 1-naphthol in the nutrient solution, 25 mL of the nutrient sample was transferred to a 50 mL volumetric flask. Then, 15 mL of methanol was added and mixed for 10 min before adding 10 mL of distilled water to make up the solution volume to 50 mL. The sample was passed through a 0.2-µm filter and kept in the freezer for no longer than 2 weeks before the HPLC analysis. All HPLC analyses were performed on an Agilent Infinity II 1260 HPLC system (Agilent Technology, Santa Clara, CA, USA) connected to a PDA detector. Chromatographic separation was carried out in a Phenomenex Luna C18 column (250 mm × 4.6 mm, Phenomenex, Aschaffenburg, Germany). An acetonitrile gradient (40–60%) was used as the mobile phases for 20 min and back to the initial condition for 10 min. The run time for each sample was 20 min, and the injection volume was 20 μL [8,23].

### 2.5. Determination of Plant Hormone in Carbaryl-Treated Plants 

The sunn hemp, sunflower, and marigold plants were grown in a hydroponic solution for 2–3 weeks and then treated with carbaryl (0, 5 or 10 mg L^−1^) for 4 days. Plant samples were cut and separated into shoots and roots, weighed, and the weights were recorded. The procedure for gibberellic acid (GA), indole-3-acetic acid (IAA), and abscisic acid (ABA) determination was modified from Trapp and Souza, Rivera and colleagues, and Sun and colleagues [24,25,26]. Each sample was ground and added with 10 mL of acetonitrile: 5% formic acid (8:2). Each mixture was poured into a 15 mL centrifuge tube and extracted using ultrasonication for 25 min. The upper solution was passed through a 0.2-μm filter and analyzed for GA, IAA, and ABA using HPLC [24,25,26]. The chemicals (GA, IAA, and ABA) were HPLC grade and purchased from TCI, Japan. The standard solutions of GA, IAA, and ABA were fresh-prepared in at least 5 concentrations (0.313–5.0 mg L^−1^). Chromatographic separation was carried out in a Poroshell 120 EC C18 column (150 mm × 4.6 mm, 4 μm, Agilent Technology, USA). Gradient acetonitrile: 0.5% formic acid was used as the mobile phase. An acetonitrile gradient (10–60%) was used for the mobile phases for 20 min with a flow rate of 0.5 mL min^−1^. The run time for each sample was 20 min, and the injection volume was 10 μL.

### 2.6. Seed Germination Experiment 

Seeds of sunn hemp, sunflower, and marigold were tested for seed germination in carbaryl solution. The experiment was modified by Goswami and colleagues, Jesus and colleagues, and Khah [27,28,29]. Seeds of uniform size were surface-sterilized in 3% sodium hypochlorite (NaOCl) for 5 min and thoroughly soaked twice using sterilized distilled water for 4 min. Then, 10 seeds were placed in a Petri dish (9 cm diameter) containing a sterilized 3-layer filter paper moistened with 10 mL of the respective treatment (5 or 10 mg L^−1^ solution of carbaryl pesticide prepared in distilled water), with 0 mg L^−1^ used as the control also using distilled water. The experiment was repeated in triplicate. All Petri dishes were incubated for 7 d in the dark at 28 ± 2 °C to promote seed germination. The emergence of root tips and shoots was determined based on their lengths, and the germination percentage of seeds was recorded [27,28,29].

### 2.7. Statistical Analysis

All statistical analyses were performed using the SPSS software. A *p*-value less than 0.05 was considered significant. Differences in average data were analyzed using One-way analysis of variance (ANOVA) and Tukey’s honestly significant difference (HSD) test [13].

## 3. Results

### 3.1. Toxicity of Carbaryl on Sunn Hemp, Sunflower, and Marigold

#### 3.1.1. Toxicity Symptoms on Plants 

The toxicity symptoms of the three flower plants are provided in Table 1 and Figure 1, Figure 2 and Figure 3. Sunn hemp showed the most toxicity symptoms, with pale-to-white spots on leaves, gray-to-dark gray roots and shrunken leaf edges (Table 1 and Figure 1, Figure 2 and Figure 3). Pale spots on the leaves of sunn hemp were observed after 4 days of cultivation for the carbaryl treatments (5 and 10 mg L^−1^, as shown in Figure 1b,c, whereas the white spots were clearly evident after cultivation for 8 and 12 days for both carbaryl treatments (5 and 10 mg L^−1^ as shown in Figure 1d–g. The shrunken leaf edge of sunn hemp was visible in the carbaryl treatments of 10 mg L^−1^ after cultivation for 4, 8, and 12 days (Figure 1c,e,g). Sunflower and marigold showed mild effects of carbaryl toxicity (Table 1 and Figure 2 and Figure 3). There were not many pale spots on the sunflower leaves initially after 4 days of cultivation for both carbaryl treatments (5 and 10 mg L^−1^), as shown in Figure 2b,c, whereas they were present on the marigold leaves initially after 4 days of cultivation at the high carbaryl concentration of 10 mg L^−1^ (Figure 3c). The shrunken leaf edges of the sunflower were clearly visible after cultivation for 8 and 12 days with the carbaryl treatment of 10 mg L^−1^, while none were evident on the marigold (Table 1). Another toxicity symptom of carbaryl detected on the plants was the change in color of their roots from white to gray. The sunn hemp had darker gray roots when exposed to carbaryl at 10 mg L^−1^ after 4 days of cultivation that turned to dark gray when exposed to carbaryl at 10 mg L^−1^ for 12 days of cultivation (Table 1). The sunflower roots started to turn gray after 8 days of cultivation with the carbaryl treatment of 10 mg L^−1^ and after 12 days of cultivation for the carbaryl treatments of 5 and 10 mg L^−1^ (Table 1 and Figure 2d–f). The marigold roots turned slightly gray only when exposed to carbaryl at 10 mg L^−1^ after cultivation for 4, 8, and 12 days (Table 1 and Figure 3e). Thus, it seemed that plant performance regarding tolerance to carbaryl toxicity was in the order: sunn hemp < sunflower < marigold. The ecotoxicity of carbaryl, as displayed by plant symptoms, may be useful as bioindicators in contaminated environments [30].

#### 3.1.2. Pigment Contents in Plants

The pigment contents after 4 days of cultivation in the three flower plants treated and not treated with carbaryl cultivation are shown in Figure 4. Sunn hemp showed the most toxicity symptoms regarding the reduction in pigment contents (Figure 4). Plants treated with carbaryl at 5 and 10 mg L^−1^ had significantly decreased total chlorophyll (77.4 and 58.0 mg kg^−1^, respectively), chlorophyll a (56.2 and 41.8 mg kg^−1^, respectively), chlorophyll b (21.2 and 15.8 mg kg^−1^, respectively), and carotenoid contents (13.7 and 10.0 mg kg^−1^, respectively) compared to the untreated plant (126.8, 84.7, 42.1, 26.4 mg kg^−1^, respectively, Figure 4). Compared to the sunn hemp, the sunflower and marigold showed less toxicity regarding pigment contents (Figure 4). Sunflower had a significantly decreased carotenoid content when exposed to carbaryl at 10 mg L^−1^ (9.7 mg kg^−1^) compared to the untreated plants (12.5 mg kg^−1^), while marigold had a significantly lower total chlorophyll content (54.1 mg kg^−1^) following exposure to carbaryl at 10 mg L^−1^ compared to the untreated plant (67.4 mg kg^−1^, *p* < 0.05, Figure 4). The results confirmed that sunn hemp was the most susceptible to carbaryl toxicity of the three plants.

#### 3.1.3. Physiological Effects of Carbaryl on Plants

Data on the relative growth rate, shoot and root lengths, leaf number, and shoot and root weights of the three plants (sunn hemp, sunflower, and marigold) are shown in Figure 5. The relative growth rates among the plants were not significantly different (Figure 5a); however, after 4 days of cultivation, sunflower had the longest shoot length at carbaryl treatments of 5 and 10 mg L^−1^ (24.75 and 26.75 cm, respectively), the highest leaf number (13.5 leaves), and the highest shoot weight (10.92 g) at the carbaryl treatments of 10 mg L^−1^ compared to the two other plants (Figure 5b,d,e). At carbaryl concentrations of 5 and 10 mg L^−1^, sunn hemp had the shortest shoot length (17.5 and 15.3 cm, respectively, Figure 5b), the lowest leaf number (8.7 and 6.7 leaves, respectively, Figure 5d), the lowest shoot weight (1.3 and 0.7 g, respectively, Figure 5e), and the lowest root weight (0.7 and 0.4 g, respectively, Figure 5f) but notably, the longest root length at the carbaryl concentration 10 mg L^−1^ (55.1 cm, Figure 5c) compared to the other two species (Figure 5). In contrast, marigold was not affected by carbaryl toxicity, and there were no significant differences in the relative growth rate, shoot and root lengths, shoot and root weights, and leaf number between the carbaryl-treated and untreated plants. Carbaryl had a clear effect on the roots and shoots of sunn hemp and sunflower (Figure 5). The physiological effect of carbaryl on sunn hemp was greater than on sunflower and marigold, respectively.

### 3.2. Accumulation of Carbaryl and its Degradate (1-Naphthol) in Flower Plants

Carbaryl and 1-naphthol accumulation in the three plants was examined in the shoots and roots (Figure 6). There were no significant differences in shoot accumulation of carbaryl among the three plant species; however, for 1-naphthol accumulation, marigold had the highest shoot accumulation when exposed to carbaryl at 5, and 10 mg L^−1^ (4.5 and 2.9 mg kg^−1^, respectively, Figure 6a) compared to the two other plants. Marigold may uptake carbaryl and/or 1-naphthol and translocate either/both to shoots or degrade the up-taken carbaryl via an internal pathway (Figure 6). Marigold had a markedly higher uptake capacity of carbaryl (194.0 mg kg^−1^) than either sunn hemp (27.8 mg kg^−1^) or sunflower (7.3 mg kg^−1^) when exposed to carbaryl at 10 mg L^−1^ but showed no uptake capacity of 1-naphthol compared to the sunn hemp and sunflower (32.1 and 10.5 mg kg^−1^, respectively, Figure 6c). The three plants showed carbaryl accumulation in the shoots and roots after 14 days of cultivation but none in the roots, suggesting plants may uptake up carbaryl and translocate it from the roots to the shoots rather than degrading carbaryl to 1-naphthol during the 4 days of exposure to carbaryl (Figure 6a–d). Notably, marigold seemed to uptake carbaryl, accumulate it, and may even have degraded it to 1-naphthol in the shoots (Figure 6). The highest translocation factor of carbaryl in marigold after 4 days of cultivation was for the carbaryl treatments of 5 and 10 mg L^−1^ (165.9 and 88.0, respectively), showing that marigold translocated more carbaryl from the roots to the shoots than the two other plants (6.7–33.0, Figure 6d). Sunn hemp and sunflower had significantly higher carbaryl accumulation levels in the roots than marigold (Figure 6b). For the carbaryl treatment at 5 mg L^−1^, marigold had a higher translocation factor than for the carbaryl treatment at 10 mg L^−1^ (Figure 6d), and for this level of carbaryl, sunflower had greater root accumulation than carbaryl at 5 mg L^−1^ (Figure 6b). Thus, marigold and sunflower could better handle the lower and higher concentrations of carbaryl treatment, respectively. Sunn hemp had the lowest translocation factor for carbaryl. It accumulated the highest level of carbaryl in roots (2.4 mg kg^−1^ for a carbaryl treatment at 5 mg L^−1^, Figure 6b) and had the highest level of uptake capacity of 1-naphthol for carbaryl treatments at 5 and 10 mg L^−1^ (13.7 and 32.1 mg kg^−1^, respectively, Figure 6c). Thus, sunn hemp may better deal with carbaryl in the roots. Although sunn hemp showed more toxicity symptoms and pigment reduction than sunflower (Table 1 and Figure 1, Figure 2, and Figure 4), sunn hemp had higher carbaryl accumulation in the roots than sunflower and marigold when exposed to carbaryl at 5 mg L^−1^ (Figure 6b) and it could uptake more 1-naphthol than sunflower or marigold when exposed to carbaryl at 5 and 10 mg L^−1^ (Figure 6c). For the carbaryl treatment at 10 mg L^−1^, sunflower had higher carbaryl accumulation in roots and a higher translocation factor than sunn hemp (Figure 6b,d). This suggested that sunn hemp could better deal with carbaryl at the lower concentrations (5 mg L^−1^) and sunflower could better deal with carbaryl at the higher concentration (10 mg L^−1^). In addition, marigold had the highest uptake capacity of carbaryl and the highest translocation factor for the carbaryl treatments at 5 and 10 mg L^−1^ (Figure 6c,d) and tended to be the carbaryl accumulator.

The comparison of the carbaryl removal efficiency levels of the three plants (Table 2) indicated that marigold had the highest TF levels at 88 compared to sunn hemp and sunflower when exposed to carbaryl at 10 mg L^−1^ after 4 days of cultivation. Sunn hemp and sunflower had similar levels of carbaryl removal efficiency after 4 days of cultivation for the carbaryl treatment at 10 mg L^−1^ (Table 2). Due to the high levels of TF, BCF (1848), and EF (1827) in marigold, there was effective carbaryl uptake by marigold and its translocation from the roots to the shoots, with low levels of carbaryl detected in the nutrient solution and better carbaryl removal efficiency after 4 days of cultivation. The marigold BCF values were above 1000, and such values can be used to indicate hyperaccumulation in a plant [31]. Thus, the present results indicated that the marigold was a hyperaccumulator for carbaryl pesticide (Table 2). However, the removal percentages of carbaryl by the three plants were not significantly different (93%, 71%, and 65% for marigold, sunflower, and sunn hemp, respectively) after 4 days of cultivation (Table 2).

### 3.3. Carbaryl Degradation to 1-Naphthol in Three Flower Plants

The carbaryl and 1-naphthol accumulation levels in sunn hemp, sunflower, and marigold after exposure to carbaryl at 5 and 10 mg L^−1^ after cultivation for 4, 8, and 12 days are shown in Figure 7. Sunn hemp showed significantly increased carbaryl accumulation in the shoots after 8 days of cultivation (from 25.1 mg kg^−1^ to 91.7 mg kg^−1^) for carbaryl treatments at 5 and 10 mg L^−1^, respectively (Figure 7a). 1-naphthol accumulation in the shoots also significantly increased after 8 days of cultivation (from 0.1 mg kg^−1^ to 0.5 mg kg^−1^) for carbaryl treatments at 5 and 10 mg L^−1^, respectively (Figure 7b). Showing the same trend, carbaryl accumulation in sunn hemp roots significantly increased after 8 days of cultivation (from 2.9 mg kg^−1^ to 22.5 mg kg^−1^) for carbaryl treatments at 5, and 10 mg L^−1^, respectively (Figure 7c) and 1-naphthol was detected (0.1 mg kg^−1^) after 8 days of cultivation for carbaryl treatment at 10 mg L^−1^. At a high concentration of carbaryl (10 mg L^−1^), sunn hemp had the highest carbaryl and 1-naphthol accumulation levels in the shoots and roots after 8 days of cultivation (Figure 7a,c), whereas sunflower accumulated significantly more carbaryl in the roots than sunn hemp after 4 days of cultivation (Figure 7c). Marigold and sunflower had the significantly highest 1-naphthol accumulation levels in the shoots and carbaryl accumulation in the roots after 4 days of cultivation at the carbaryl treatments at 10 mg L^−1^ (Figure 7a,c). After 8 days and 12 days of cultivation, these levels of carbaryl accumulation in the shoots and roots decreased. In addition, marigold had the significantly highest 1-naphthol accumulation in the shoots when treated with carbaryl at 5 mg L^−1^ after 8 days of cultivation and significantly increased the 1-naphthol accumulation in the shoots when treated with carbaryl at 10 mg L^−1^ after 12 days of cultivation (Figure 7b). In addition, 1-naphthol was not detected in the roots of either sunflower or marigold, suggesting the good ability of marigold to accumulate a large amount of carbaryl initially and to then degrade it to 1-naphthol and store it in the shoots afterward. There was detectable 1-naphthol only in the roots of sunn hemp after 8 days of cultivation with 10 mg L^−1^ of carbaryl exposure (0.09 mg kg^−1^). After high carbaryl exposure (10 mg L^−1^), the roots of sunn hemp could accumulate a significantly high amount of carbaryl after 8 days of cultivation (22.5 mg kg^−1^, Figure 7c) concurrent with a lower amount of 1-naphthol at the same time. This may have been due to the uptake of carbaryl with its degradation, 1-naphthol, from the nutrient solution, and/or carbaryl degradation to 1-naphthol in the roots of sunn hemp. Compared to marigold and sunflower, sunn hemp is a legume with root colonization ability. This might have affected carbaryl degradation to 1-naphthol in the rhizosphere zone in the nutrient solution.

### 3.4. Effect of Carbaryl on Internal Plant Hormone

The levels of gibberellic acid (GA), indole-3-acetic acid (IAA), and abscisic acid (ABA) in sunn hemp, sunflower, and marigold treated with and without carbaryl (0, 5, or 10 mg L^−1^) in the plant shoots and roots during plant cultivation at day 0 (commencement) and at 4 days of cultivation. The results showed that at the commencement of cultivation (day 0), there were no detectable plant hormones (GA, IAA, or ABA); however, after 4 days of cultivation, sunn hemp roots had very low levels of GA in both the untreated and treated plants with carbaryl at 5 mg L^−1^. In sunn hemp treated with the higher level of carbaryl (10 mg L^−1^), the roots contained IAA at 10.8 mg kg^−1^. The detectable plant growth hormones, The presence of GA and IAA in sunn hemp may have been the reason for its great ability to accumulate carbaryl in the roots when exposed to carbaryl at 5 and 10 mg L^−1^ (Figure 6b and Figure 7c). Furthermore, the physiological change to longer sunn hemp roots for the carbaryl treatment at 10 mg L^−1^ after 4 days of cultivation (Figure 5c) was possibly related to its hormone (IAA) level. No GA, IAA, or ABA were detected in marigold shoots and roots from commencement or after 4 days of cultivation; indeed, this plant is a hyperaccumulator for carbaryl pesticide. ABA was detected in sunflower shoots at increased levels from 16.9 mg kg^−1^ for the carbaryl treatment at 5 mg L^−1^ to 25.3 mg kg^−1^ in the carbaryl treatment at 10 mg L^−1^ after 4 days of cultivation. ABA is a plant-growth-inhibiting hormone that might have paused the carbaryl accumulation process in the shoots of sunflower for the carbaryl treatment at 10 mg L^−1^ after 4 days of cultivation, although it could produce more biomass (as indicated by the shoot length and the shoot and root weights in Figure 5b,d–f) and the high carbaryl accumulation in the roots (Figure 6b and Figure 7c).

### 3.5. Effect of Carbaryl on Seed Germination

Seeds of sunn hemp, sunflower, and marigold were tested for their germination percentage and the shoot and root lengths of the seed sprouts under carbaryl stress at concentrations of 0, 5, or 10 mg L^−1^ (Figure 8). The germination percentages of the sunn hemp, sunflower, and marigold seeds were 100, 62.5 and 100%, respectively. The condition test of the 10 mL solution without carbaryl was best for the sunn hemp and marigold seeds, while the 5 mL solution had a better germination percentage of sunflower seed. With carbaryl stress of 5 and 10 mg L^−1^, the germination percentages of sunn hemp were both 100% and for marigold were 100% and 70.8%, respectively. Sunn hemp had the best seed germination with a high concentration of carbaryl at 10 mg L^−1^; however, dark gray root sprouts were observed. Due to the hyperaccumulation ability of marigold and the leguminous ability of carbaryl accumulation with sunn hemp, their lengths of shoot and root sprouts were investigated further (Figure 8). The results showed that sunn hemp had reduced shoot sprout lengths (3.7 and 3.5 cm) and root sprout lengths (0.9 and 0.7 cm) when treated with carbaryl at 5 and 10 mg L^−1^, respectively, compared to the untreated lengths (12.8 cm for shoots and 2.7 cm for roots, Figure 8). There were no significant differences in the lengths of shoots or root sprouts for marigold between the carbaryl treatments at 5 and 10 mg L^−1^ and the untreated carbaryl (Figure 8). Carbaryl could reduce the germination percentage of the marigold seeds exposed to high carbaryl stress at 10 mg L^−1^ (70.8% germination); however, this high level of carbaryl did not significantly affect the shoot and root lengths of the marigold sprouts (Figure 8). Plant hormones may have been the cause of the effect of carbaryl on seed germination. The internal plant hormone should be determined further for clarification.

## 4. Discussion

### 4.1. Phytoextraction of Carbaryl by the Flower Plants

Phytoremediation (the use of plants for the treatment of contaminated soil, sediment, and water bodies) is a technology that includes phytostabilization, phytoextraction, phytovolatilization, and phytofiltration [32]. In the present study, three flower plants were evaluated for their phytoextraction ability based on the removal of carbaryl pesticide in a hydroponic system. Marigold performed well for carbaryl phytoextraction with the greatest shoot accumulation and uptake compared to the other two species (Figure 6a,c). The phytostabilization ability of marigold has been reported for Cd by Thongchai and colleagues [19]. Five different cultivars of marigold were tested, and the Babuda and Sunshine cultivars grown with pig manure produced the greatest biomass (22.3 and 34.1%, respectively), the greatest Cd accumulation (1342.8 and 1261.8 mg plant^−1^, respectively) and flower production [19]. The high tolerance to Cd-induced toxicity by activation of its antioxidative defense system was a chosen characteristic for the plant’s application in a pollution-contaminated environment [19]. In the present study, different marigold cultivars should be evaluated for their level of phytoextraction ability further in both hydroponic and soil systems for further use in field application. 

The phytoextraction ability of other leguminous plants has been reported, showing the same trends as for sunn hemp in the present study [12]. Lupine (*Lupinus angustifolius*) was shown to have the phytoextraction ability for carbaryl in a hydroponic system [12]. In that study, when plants were exposed to 10 and 50 mg L^−1^ of carbaryl for 16 days, the carbaryl concentration was as high as 48% and 57%, respectively, that was degraded and/or bound in an irreversible manner with plant material. Partial carbaryl concentrations of 2% were found in the roots and also in the shoots of lupine exposed to a carbaryl concentration of 10 mg L^−1^ [12]. Furthermore, their seed germination test showed that lupine could germinate under carbaryl stresses of 10 and 50 mg L^−1^ in a water solution with germination indices of 4.6 and 4.4, respectively, compared to the non-stressed control (4.9). A seed germination experiment is important as it reveals the survival of the plant after the treatment and the toxicity of carbaryl. In the present study, sunn hemp had germination percentages of 100% and 70.8% when exposed to carbaryl concentrations of 5 and 10 mg L^−1^, respectively. It could accumulate carbaryl in the shoots and roots as high as 73.1 and 6.3 mg kg^−1^, respectively, when exposed to a carbaryl concentration of 10 mg L^−1^ (Figure 7a,c), which suggested that sunn hemp performed survived well under carbaryl stress and could be useful for the remediation of carbaryl contamination.

In the present study, sunflower also has high removal percentage of carbaryl (71%, Table 2). Sunflower has been reported to accelerate the degradation of the carbamate pesticide carbofuran [3]. The planted soil had a degradation rate constant of 0.31 day^−1^ with a half-life of 2 days, whereas non-planted soil had a degradation rate constant of 0.24 day^−1^ with a half-life of 3 days [3]. The half-life of a pesticide is affected by the pH, soil type, temperature, moisture content and microbial population. Pesticides are bound in a manner so that they are no longer available to most biological processes, and ordinary chemical extraction procedures are often considered "degraded" [3,12]. Many plants can release molecules from the roots as exudates that could enhance pesticide degradation [3,12]. In the present study, sunflower has significantly high carbaryl accumulation in the roots (8.7 mg kg^−1^, Figure 7c) when exposed to a carbaryl concentration of 10 mg L^−1^ at 4 days of cultivation. The presence of ABA might be involved in the plant response to limit shoot accumulation of carbaryl and/or possibly in the root response to carbaryl contamination in a nutrient solution.

### 4.2. Carbaryl Degradation to 1-Naphthol in Nutrient Solution and Its Toxicity in Plants

After 4 days of cultivation without plant nutrient solution, adding carbaryl at 5 and 10 mg L^−1^ produced baseline degradation levels to 1-naphthol in similar ranges of 21.1–76.1% and 28.8–65.1% degradation, respectively) after every 4 days (4 days, 8 days, and 12 days of cultivation). This may have been due to reactive oxygen species (ROS), including singlet oxygen (^1^O_2_) and hydroxyl radicals (*OH), being photogenerated in the water and playing important roles in the indirect photolysis of carbaryl, resulting in carbaryl degradation to 1-naphthol [33]. In the present study, there was the possibility of plants taking up both carbaryl and 1-naphthol through their roots and performing carbaryl removal from contaminated solution (Figure 6 and Figure 7). Systemic ROS inside the plants can be induced and accumulated in response to light stress, wounding, and pathogen infection [34]. The major ROS production sites and processing pathways in plants are in the chloroplast, apoplast, mitochondria, peroxisomes, cytosol, cell walls, and membranes of plant cells [35]. ROS-associated genes are involved in the control of gametophyte and flower development, such as determining fertility, the development of gametophytes, and petal development [35]. During plant growth, the effects of carbaryl in reducing the pigment contents of chlorophyll and carotenoid were shown in sunn hemp, sunflower, and marigold (Figure 4). Although these plants had constant relative growth rates (Figure 5a), carbaryl affected their physiological appearances, such as the shoot and root weights and lengths (Figure 5).

### 4.3. Plant Responses to Carbaryl Accumulation

Sunn hemp showed high accumulation levels of carbaryl and 1-naphthol in the shoots and roots after 8 days of cultivation for the carbaryl treatment at 10 mg L^−1^, with greater toxicity symptoms than for sunflower or marigold (Figure 7a–c). Under low carbaryl stress (5 mg L^−1^), sunn hemp may accumulate carbaryl and detoxify it in the roots (Figure 6b and Figure 7c). In general, sunn hemp symbiosis with beneficial microbes in root nodules may have promoted plant growth by siderophore synthesis, phosphorus solubilization, and the production of plant growth hormones, such as IAA [36]. Siderophore biosynthesis in microorganisms is triggered by intracellular iron deficiency and is secreted for scavenging iron to microbes, including the host plant [37]. The legume nodule biomass can influence plant growth, yield, and nitrogen fixation and is strongly correlated with available phosphorus [38]. Thus, sunn hemp, as a legume, possibly facilitates the roots absorbing phosphate and/or nutrient for plant growth and facilitates the phytoextraction of carbaryl. In the present study, the amount of IAA was as high as 10.8 mg kg^−1^ when sunn hemp was treated with carbaryl at 10 mg L^−1^. This result was in agreement with Berg and colleagues and Egamberdieva and colleagues, who reported that the nodulating bacterium *Stenotrophomonas rhizophila* in sunn hemp could produce phytohormones and IAA; furthermore, the bacterial IAA increased the root surface area resulting in a greater surface area for water and nutrient uptake from the environment that had a direct effect on plant growth under stress and non-stress conditions [39,40]. In the present study, sunn hemp had the longest root length when treated with carbaryl at 10 mg L^−1^ after 4 days of cultivation, with the IAA production comparable to the other two plant species (Figure 5c), and it could accumulate carbaryl and 1-naphthol at high levels in its shoots and roots after 8 days of cultivation (Figure 7a,c). The carbaryl detoxification mechanism of sunn hemp might involve plant-microbe interactions. In addition, sunn hemp can grow in a variety of soil types and with variable rainfall in tropical or subtropical environments [41]. The germination percentage (100%) of sunn hemp seed subjected to carbaryl contamination levels at 5 and 10 mg L^−1^ showed that it has more potential to address succession in a stressful environment. It is readily used as a rotational crop in tropical regions with cash crops such as rice, cotton, and corn [41]. The proposed model of sunn hemp response to carbaryl accumulation can be summarized as follows. When sunn hemp was treated with low levels or without carbaryl (0 and 5 mg L^−1^), it produced GA, and when treated with a high level of carbaryl (10 mg L^−1^), it produced IAA. The IAA phytohormone induced elongation in the roots, and they accumulated carbaryl and 1-naphthol. Apart from this, some amounts of carbaryl and 1-naphthol were translocated and accumulated in the shoots.

The results from the present study showed that after 4 days of cultivation with carbaryl toxicity at 5 and 10 mg L^−1^, sunflower could induce an internal plant hormone (ABA) in its shoots (16.9 and 25.3 mg kg^−1^, respectively). Although it accumulated carbaryl well in the roots after 4 days of cultivation when exposed to a high concentration of carbaryl (10 mg L^−1^), its shoots produced a high level of ABA compared to the other two plant species. ABA, a stress phytohormone, plays an important role in protecting the plant from various biotic and abiotic stresses in the environment [42]. Plant internal ROS can trigger ABA, with the ABA inducing various defense signaling systems to protect the plant from adverse effects. This may have resulted in the low carbaryl accumulation in the shoots in contrast to the high carbaryl accumulation in the roots of the sunflower after 4 days of cultivation when exposed to carbaryl at 10 mg L^−1^; in addition, the sunflower had a greater shoot and root weights (Figure 5e,f and Figure 6b). The lack of evidence of 1-naphthol in the sunflower might have been due to a further detoxification mechanism. Carbaryl degradation to 1-naphthol may have induced a ROS signaling response by the plant, which triggered ABA production when the plant was exposed to carbaryl at 10 mg L^−1^. ABA inhibits GA synthesis and upregulates ABA signaling pathways that inhibit carbaryl accumulation, translocation, and degradation to 1-naphthol in sunflower shoots [36].

Marigold was indicated as a hyperaccumulator in the present study (Table 2). The toxicity symptoms and reduction in the pigment contents in marigold due to the carbaryl were not as apparent as they were in sunn hemp (Table 1 and Figure 1, Figure 3 and Figure 4). However, the seed germination percentage of marigold, when treated with carbaryl at 10 mg L^−1^, was lower (70.8%) than for sunn hemp (100%). In general, plant hormones, such as GA, are synthesized and stimulate seed germination, while ABA (if present due to induction by accumulated ROS) can promote ABA signaling pathways that inhibit the germination process [35]. Carbaryl stress may cause ROS in the plant, and then the ROS could induce the ABA or inhibit the GA pathways during seed germination in marigold or sunn hemp. In the present study, the proposed model of marigold response to carbaryl accumulation could be that marigold can take up carbaryl and 1-naphthol in its roots, accumulate them there, and then translocate them for further accumulation in its shoots. 

## 5. Conclusions

Marigold showed fewer toxicity symptoms due to carbaryl, having the highest BCF (1848), where a value of more than 1000 indicates a hyperaccumulator (Table 2). Thus, marigold had the highest phytoremediation ability for carbaryl removal compared to sunn hemp and sunflower. In addition, marigold had the highest TF, suggesting plant uptake and translocation of carbaryl that had accumulated in the shoots (Figure 6d). Carbaryl degradation to 1-naphthol in marigold may have occurred due to the accumulation of degraded 1-naphthol in its shoots (Figure 7b). Sunflower showed fewer toxicity effects to carbaryl and could accumulate carbaryl in its roots at a higher level in a shorter period than sunn hemp when exposed to the higher carbaryl concentration of 10 mg L^−1^ (after 4 days of cultivation following carbaryl exposure, Figure 6b and Figure 7c). Sunn hemp could accumulate carbaryl in the shoots better when exposed to the higher carbaryl concentration of 10 mg L^−1^ after 8 days of cultivation (Figure 7a), with more toxicity symptoms apparent. Although sunn hemp can handle carbaryl at a low level of contamination, it is a legume that can amend soil fertility with nitrogen fixation. Thus, at a low level of carbaryl contamination, sunn hemp may be good for use as an accumulator for carbaryl removal, whereas, for a high level of carbaryl contamination, marigold could be used as a phytoremediator with hyperaccumulation ability for carbaryl removal.

## Figures and Tables

**Figure 1 ijerph-19-16482-f001:**
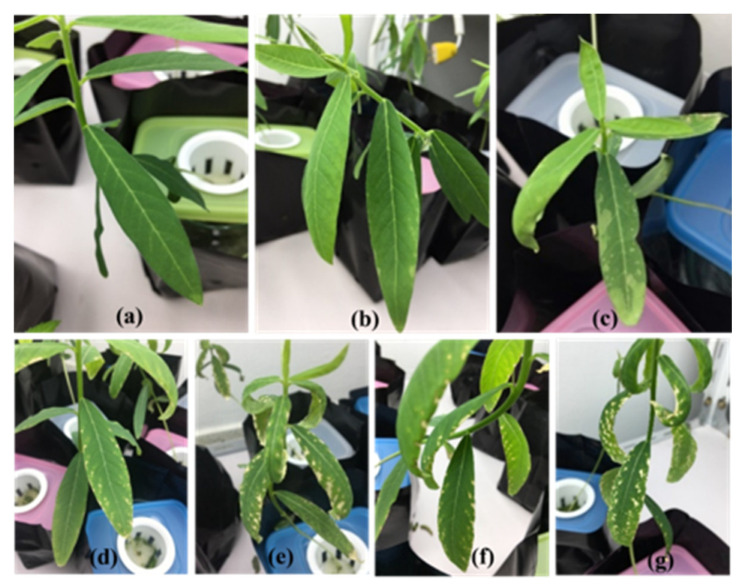
Toxicity in sunn hemp after 4 days of cultivation for carbaryl treatments at 0 mg L^−1^ (**a**), 5 mg L^−1^ (**b**), 10 mg L^−1^ (**c**), after 8 days of cultivation for carbaryl at 5 mg L^−1^ (**d**) and 10 mg L^−1^ (**e**), after 12 days of cultivation for carbaryl at 5 mg L^−1^ (**f**), and 10 mg L^−1^ (**g**).

**Figure 2 ijerph-19-16482-f002:**
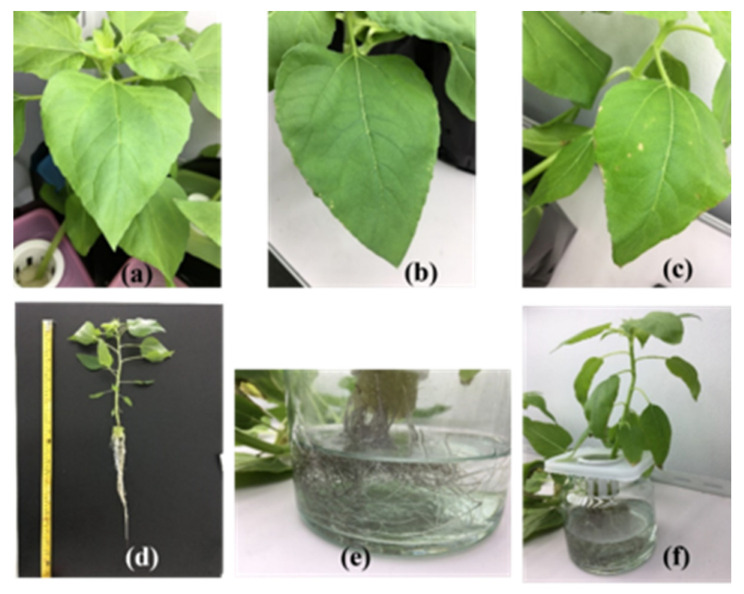
Toxicity in sunflower after 4 days of cultivation for carbaryl treatments at 0 mg L^−1^ (**a**), 5 mg L^−1^ (**b**), and 10 mg L^−1^ (**c**), after 12 days cultivation (control) (**d**), and gray roots for carbaryl at 10 mg L^−1^ (**e**,**f**).

**Figure 3 ijerph-19-16482-f003:**
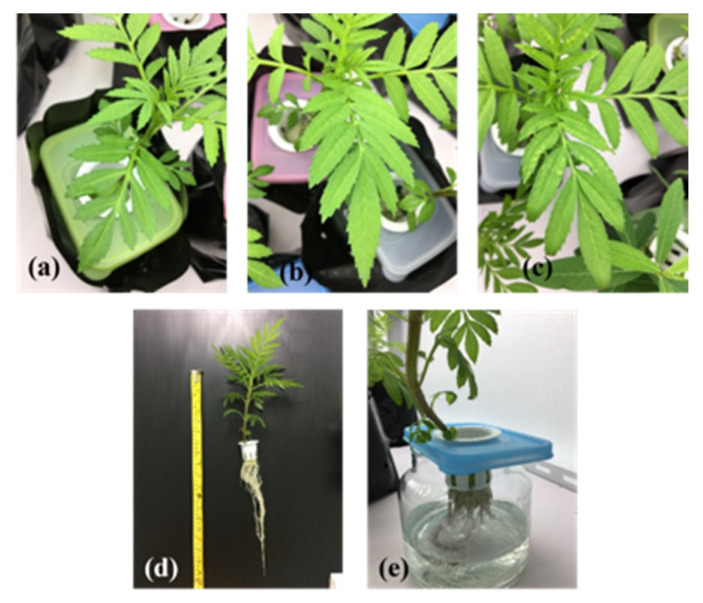
Toxicity in marigold after 4 days of cultivation for carbaryl treatments at 0 mg L^−1^ (**a**), 5 mg L^−1^ (**b**), and 10 mg L^−1^ (**c**), after 12 days of cultivation for carbaryl at 10 mg L^−1^ (control) (**d**) and gray roots (**e**).

**Figure 4 ijerph-19-16482-f004:**
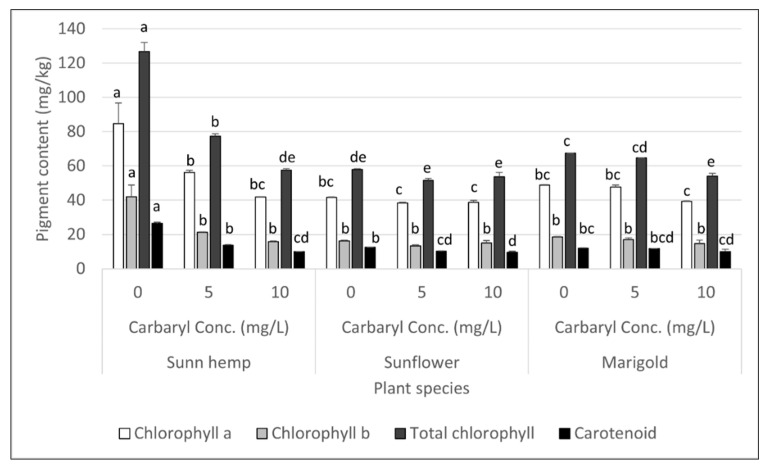
Effect of carbaryl on pigment contents of plants after 4 days of cultivation for carbaryl treatments at 0, 5 or 10 mg L^−1^. Note: a, b, c, d, and e indicate significant differences based on ANOVA and Tukey’s HSD test. The data shown are mean and SD.

**Figure 5 ijerph-19-16482-f005:**
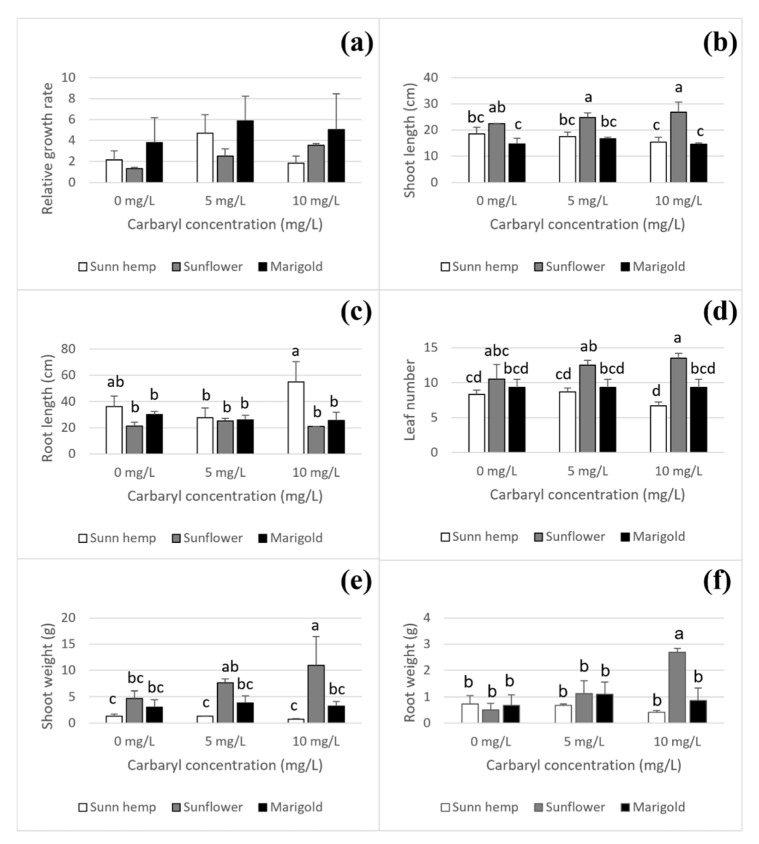
Effect of carbaryl on plant physiology after 4 days of cultivation for carbaryl treatments at 0, 5 and 10 mg L^−1^; relative growth rate (**a**), shoot length (**b**), root length (**c**), leaf number (**d**), shoot weight (**e**), and root weight (**f**). Note: a, b, c, and d indicate significant differences based on ANOVA and Tukey’s HSD test. The data shown are mean and SD.

**Figure 6 ijerph-19-16482-f006:**
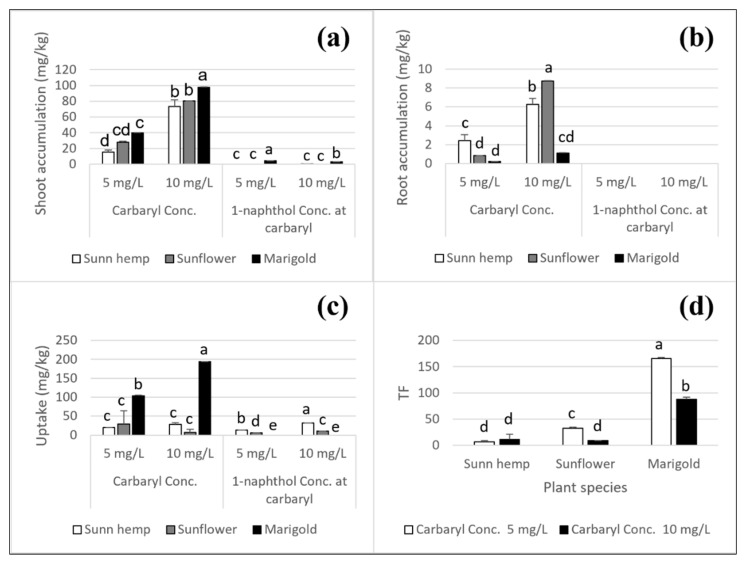
Carbaryl and 1-naphthol accumulation in plants after 4 days of cultivation for carbaryl treatments at 5 and 10 mg L^−1^; shoot accumulation (**a**), root accumulation (**b**), uptake capacity (**c**), and translocation factor (**d**). Note: a, b, c, d, and e indicate significant differences based on ANOVA and Tukey’s HSD test. The data shown are mean and SD.

**Figure 7 ijerph-19-16482-f007:**
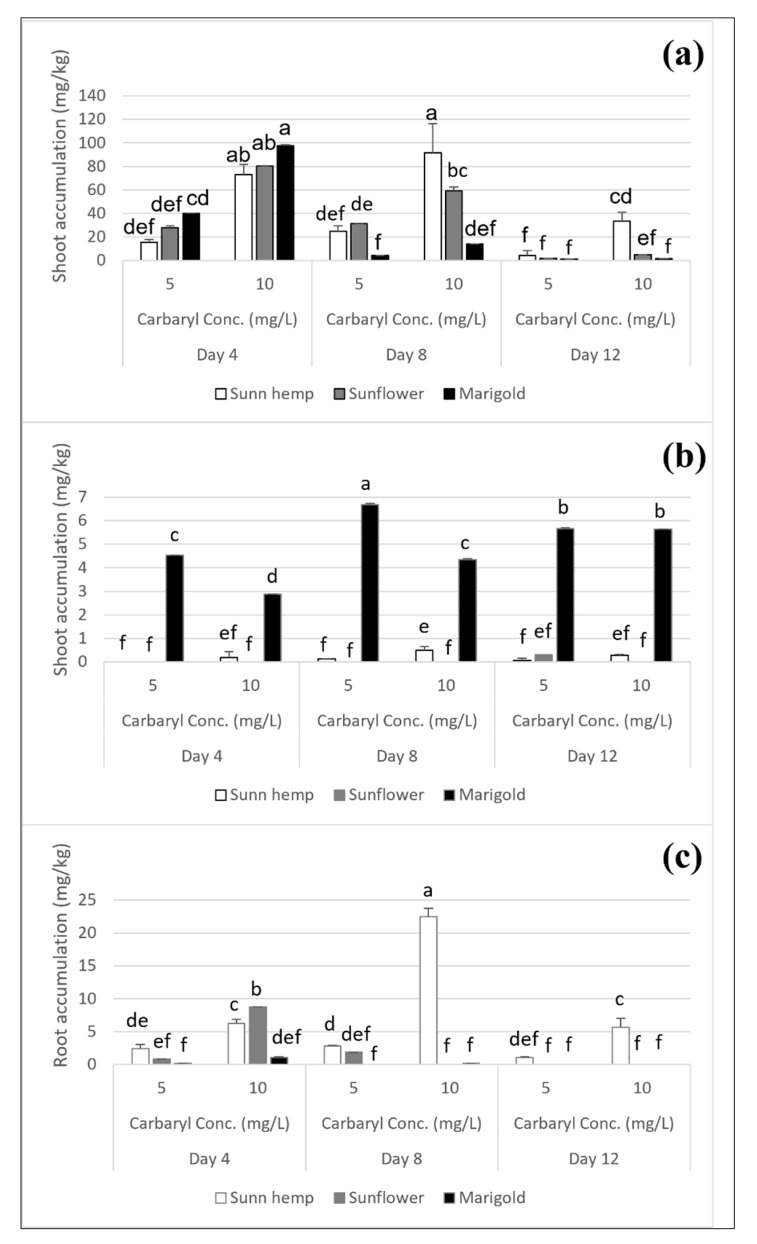
Carbaryl degradation in shoots and roots of sunn hemp, sunflower, and marigold after 4, 8, and 12 days of cultivation; carbaryl in shoots (**a**), 1-naphthol in shoots (**b**); and carbaryl in roots (**c**). Note: a, b, c, d, e, and f indicate significant differences based on ANOVA and Tukey’s HSD test. The data shown are mean and SD.

**Figure 8 ijerph-19-16482-f008:**
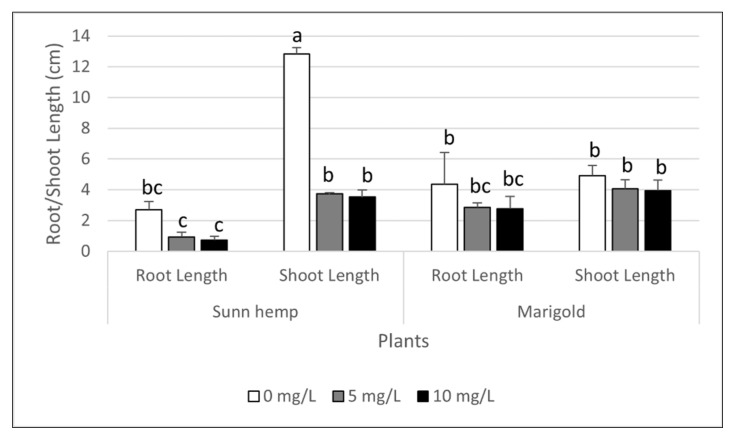
Seed germination test for the length of shoot and root of sunn hemp and marigold exposed to carbaryl concentrations of 0, 5 and 10 mg L^−1^. Note: a, b and c indicate significant differences based on ANOVA and a Tukey HSD test. The data shown are mean and SD.

**Table 1 ijerph-19-16482-t001:** Toxicity symptoms were observed in plants exposed to carbaryl for 4, 8, and 12 days ^1^.

Plant Species	Time (Days)	Carbaryl Conc. (mg L^−1^)	Toxicity Symptom		
			Pale-to-White Spots on Leave	Gray-to-Dark-Gray Root	Shrunken Leave Edge
Sunn hemp	4	5	−	−	−
		10	+	+	+
	8	5	++	+	−
		10	++	+	+
	12	5	++	+	−
		10	+++	++	+
Sunflower	4	5	−	−	−
		10	−	−	−
	8	5	−	−	−
		10	−	+	+
	12	5	−	+	−
		10	−	+	+
Marigold	4	5	−	−	−
		10	+	+	−
	8	5	−	−	−
		10	+	+	−
	12	5	−	−	−
		10	−	+	−

Note: ^1^ Effects of carbaryl were compared on leaves and roots; +: mild effect with >80% of total chlorophyll; ++: moderate effect with >60% of total chlorophyll; +++: severe effect with <40% of total chlorophyll; −: no effect observed.

**Table 2 ijerph-19-16482-t002:** Carbaryl removal parameters in flower plants after 4 days of cultivation for carbaryl treatment at 10 mg L^−1^.

Plant Species	Carbaryl Removal Factor			
	BCF	TF	EF	Removal Percentage
Sunn hemp	9.51 ± 7.43	11.21 ± 10.50 ^b^	8.69 ± 7.53	65.11 ± 49.44
Sunflower	15.59 ± 0.04	9.20 ± 0.03 ^b^	14.12 ± 1.33	71.02 ± 50.19
Marigold	1847.63 ± 859.67	88.00 ± 4.01 ^a^	1826.63 ± 849.06	93.03 ± 12.74

Note: a and b indicate significant differences based on ANOVA and Tukey’s HSD test. The data shown are mean and SD. BCF; Bioconcentration factor, TF; Translocation factor, EF; Enrichment factor.

## Data Availability

Environmental Protection Agency, USA at https://www.epa.gov; (accessed on 6 June 2021). Department of Agriculture, Maryland, USA at https://www.aphis.usda.gov (accessed on 6 June 2021; World Health Organization Press, Geneva, Switzerland at https://cdn.who.int (accessed on 6 June 2021; Ministry of Agriculture and Cooperatives, Thailand at https://www.acfs.go.th (accessed on 6 June 2021).

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
