# Peer review of "Phytoextraction Potential of Sunn Hemp, Sunflower, and Marigold for Carbaryl Contamination: Hydroponic Experiment"

_ijerph, 2022, doi:10.3390/ijerph192416482_

Round 1

Reviewer 1 Report

Dear authors, congratulations for the paper, I have some suggestions to help:

Keywords should be different from the title

Methods: I missed a work chart showing illustrating the steps of the work.  

Insert a paragraph explaining what the Sunn hemp (Crotalaria juncea L.), sunflower (Helianthus annuus L.), and marigold (Ta-107 setes sp.) are important to study and were selected 

Is it possible to compare your study with other and equal or different conditions? I missed a comparison in the discussions section. 

References: DOI numbers are missing 

Author Response

Response to Reviewer 1

Phytoextraction potential of sunn hemp, sunflower and marigold for carbaryl contamination: Hydroponic experiment

Response to reviewer’s comment

The manuscript entitled “Phytoextraction potential of sunn hemp, sunflower and marigold for carbaryl contamination: Hydroponic experiment” has been revised according to reviewer’s comment and suggestion. We are appreciated for your kind comment and suggestion. The revised manuscript file has been rechecked English proofread and combined in one file. The edited texts have been in “Track Changes” and our response has been as point by point as follow.

Reviewer’s comment

Authors’ response

1.     Keywords should be different from the title

1.     The keywords of manuscript have been corrected by changing from sunn hemp; sunflower; marigold; bioaccumulation; carbaryl to flower plants; carbaryl; bioaccumulation; toxicity.

2.     Methods: I missed a work chart showing illustrating the steps of the work.

2.     The detail steps of work in methods of plant cultivation, toxicity study, determination of carbaryl, 1-naphthol, plant hormones (gibberellic acid; GA, indole-3-acetic acid; IAA, and abscisic acid; ABA), and seed germination tests were added in Materials and Methods section.         

3.     Insert a paragraph explaining what the sunn hemp (Crotalaria juncea L.), sunflower (Helianthus annuus L.), and marigold (Ta-107 setes sp.) are important to study and were selected.

3.     In this study many kinds of flower plants were screened. The result shown that tropical plants; sunn hemp (Crotalaria juncea L.), sunflower (Helianthus annuus L.), and marigold (Tagetes erecta) have high carbaryl removal and show tolerance. The inserted paragraph explained why these flower plants were selected to study has been added in Introduction section.

4.     Is it possible to compare your study with other and equal or different conditions? I missed a comparison in the discussions section.

4.     The comparison of our study with others has been added in the Discussion section.

5.     References: DOI numbers are missing.

5.     The reference DOI and/or links have been added in the Reference section.

Reviewer 2 Report

Overall speaking, the manuscript is well written however the author should consider the following points before considering to be published your paper in “International Journal of Environmental Research and Public Health” as follows:

1. Actually, the content of discussion is not match to the research objective of the manuscript. For example, the authors tried to explain ROS in details however I cannot find any experimental results about ROS (line 428-447). Instead, the authors just measure the RGR, BCF, TF, EF and simply seed germination test. Therefore, the authors should discuss the results accordingly instead of writing other topics in an attempt to explain the current results in the manuscript.

2. In line 86-92, there are many types of phytoremediation such as phytoextraction, phytostabilization, rhizofiltration….etc. However, the authors explained generally about phytoremediation. Please choose ONE type of phytoremediation and review its literature in detail accordingly.

3. In line 98, “…….little evidence report on using flower plants to remediate carbaryl residue in a carbaryl-contaminated environment….”. However, I found several publications related to the study topic as follows:

Setiadi et al. (2022). The mixture of agricultural pesticides and their impact on populations: bioremediation strategies. Emerging Contaminants in the Environment

Challenges and Sustainable Practices. Chapter 20 - The mixture of agricultural pesticides and their impact on populations: bioremediation strategies, Page 511-546.

Bano et al. (2020). Biosensors and Bioremediation as Biotechnological Tools for Environmental Monitoring and Protection. International Journal of Current Microbiology and Applied Sciences, 9, Page 3406-3425.

Abbasi T and Abbasi SA. (2010). Factors which facilitate waste water treatment by aquatic weeds–the mechanism of the weeds' purifying action. International journal of Environmental Studies 67, Pages 349-371.

Please CHECK carefully and list the detailed literature review about the related topics in the manuscript.

4. Line 112 to 130: In line 117, The experiment condition is not clear. For example: What is the venue condition or background information to perform experiment? Is it in the laboratory? the greenhouse? the incubator? The growth chamber? In line 118, 10h: 14h, which one is referring to light cycle or dark cycle? In line 125, what kind of solvent used for plant pigment extraction? Is there any standard method such as USEPA?

5. In line 153 to 162: Is there any quality assurance and quality control procedure? Have you used standard reference materials (e.g. from NIST) to check the recovery during analysis? What is the percentage of recovery such as 90%-110%?

6. In ine 164 to 170: What is the stage of plant used for experiment? Small plant germinated from the seed? Seedling? Mature plant? Plant sampled from the field? Is it pure breeding? Is there any genetic sequece of the plants for testing? All these factors will alter the experimental results and also the conclusion of the research. Please elaborate.

7. As I found that the seed germination test should follow USEPA method EPA/600/D-89/109 (NTIS PB90113184), 1989. Which method should be referred for your experiment?

8. In table 1, the toxicity symptom cannot reflect the real situation of the plant exposure to the toxicant. It cannot estimate it quantitatively by using “+” or “-“. Therefore, the author should measure it in a scientific way such as the total surface area of infected tissue in the leaf and compare the data using statistics so as to elminiate the bias of observing the symptom subjectively.

9. In the experimental design, is there any positive and negative control of experiment? I cannot draw any conclusion based on the cirrent data without comapring to the control treatment. Besides, the decreased concentration of carbaryl doesn’t mean that such contaminant was absorbed by the plant. The toxicant maybe degraded by the soil microorganisms or by means of photodegradation to become derivatives. Therefore, the author should provide strong evidence to support the fate of carbaryl after experiment (both in root, stem and leave of plants and hydroponic solution) otherwise the data in this current findings is unreliable.

10. Back to the basic concept, can carbaryl TOTALLY soluble in water? Can the author tell all the readers about the way to solve the problem relating to the solubility of non-polar organic compound into polar solvent such as water during experiment??

11. In figure 4 and 6, why don’t analyse your data in each species separately instead of mixing all data for anaysis using ANOVA? Suprisingly, there is NO letter showing significant difference in figure 5a!

12. In line 371 to 374, you haven’t show the data of GA, IAA, ABA concentration after experiment in the manuscript. However, the authors tried to explain the experimental results based on GA, IAA, ABA. Therefore, I think author should discuss mainly on the experimental results instead of discussing hypothetial data.

13. In figure 7b and figure 7c, is there any trend to show for your experiment? I found there is the same problem as in point 11. Please explain it clearly to the readers about the experimental results related to your current research.

14. In line 458, based on my understanding, there are significant effect on the plant  with the root legume colonization in term of nitrogen absorption rather than phosphorus absorption. Please specifiy the TYPES and SPECIES of legume as well as the status of infection so that it is more specific to absorb phosphate for the plant growth and factilitate phytoextraction.

15. The author should delete figure 9 because the content of figure is totally out of the scope for the current research theme.

To conclude, the author should address the problems: discuss the current results and elaborate clearly in the manuscript about the concepts, data interpretation, findings evaluation and possible application of such technology in the real situation.

Author Response

Response to Reviewer 2

Phytoextraction potential of sunn hemp, sunflower and marigold for carbaryl contamination: Hydroponic experiment

Response to reviewer’s comment

The manuscript entitled “Phytoextraction potential of sunn hemp, sunflower and marigold for carbaryl contamination: Hydroponic experiment” has been revised according to reviewer’s comment and suggestion. We are appreciated for your kind comment and suggestion. The revised manuscript file has been rechecked English proofread and combined in one file. The edited texts have been in “Track Changes” and our response has been as point by point as follow.

Reviewer’s comment

Authors’ response

1.     Actually, the content of discussion is not match to the research objective of the manuscript. For example, the authors tried to explain ROS in details however I cannot find any experimental results about ROS (line 428-447). Instead, the authors just measure the RGR, BCF, TF, EF and simply seed germination test. Therefore, the authors should discuss the results accordingly instead of writing other topics in an attempt to explain the current results in the manuscript.

1.     The highlights have been rewritten. The experimental results of phytoextraction of carbaryl by three plants has been discussed in Discussion section.

2.     In line 86-92, there are many types of phytoremediation such as phytoextraction, phytostabilization, rhizofiltration….etc. However, the authors explained generally about phytoremediation. Please choose ONE type of phytoremediation and review its literature in detail accordingly.

2.     Phytoextraction has been explained in detail and added in the last paragraph in Introduction section.

3.     In line 98, “…….little evidence report on using flower plants to remediate carbaryl residue in a carbaryl-contaminated environment….”. However, I found several publications related to the study topic as follows:…..

3.     The references follow your kind suggestion have been used in Introduction and cited.

4.     Line 112 to 130: In line 117, The experiment condition is not clear. For example: What is the venue condition or background information to perform experiment? Is it in the laboratory? the greenhouse? the incubator? The growth chamber? In line 118, 10h: 14h, which one is referring to light cycle or dark cycle? In line 125, what kind of solvent used for plant pigment extraction? Is there any standard method such as USEPA?

4.     All of the experiment were performed at the same cultivating condition in laboratory under light setting at 10,000 lux, and photoperiod of 10 h:14 h interval at 28+2oC. The plant cultivation procedure has been explained in detail in Material and Methods section.

Chlorophyll extraction was performed by using acetone extraction. The study of toxicity symptoms and pigment extraction procedures have been added in detail. It is the modification method from standard method which we use in our previous studied.

5.     In line 153 to 162: Is there any quality assurance and quality control procedure? Have you used standard reference materials (e.g. from NIST) to check the recovery during analysis? What is the percentage of recovery such as 90%-110%?

5.     We use standard method of carbaryl measurement and setting the stand curve for analysis. Please see supplement data for HPLC standard curve of carbaryl and 1-naphthol. The linear correlation for calculations of carbaryl and 1-naphthol concentrations are with R2>0.998.

The sources of chemical purchased, and carbaryl and 1-naphthol extraction procedures have been added in detail.

6.     In line 164 to 170: What is the stage of plant used for experiment? Small plant germinated from the seed? Seedling? Mature plant? Plant sampled from the field? Is it pure breeding? Is there any genetic sequence of the plants for testing? All these factors will alter the experimental results and also the conclusion of the research. Please elaborate.

6.     A starting biomass of three plants were of 23–45 g L-1 which were germinated from the seed we grow were use in this experiment.

There is no genetic sequence of the plants for testing. The different plant cultivars that maybe affect the test results have been discussed in Discussion section.

7.     As I found that the seed germination test should follow USEPA method EPA/600/D-89/109 (NTIS PB90113184), 1989. Which method should be referred for your experiment?

7.     Seed germination was performed from the method according to the Reference number 27-29. The results of seed germination in this study could be comparable with other research. We have added and discussed in detail in Discussion section.

8.     In table 1, the toxicity symptom cannot reflect the real situation of the plant exposure to the toxicant. It cannot estimate it quantitatively by using “+” or “-“. Therefore, the author should measure it in a scientific way such as the total surface area of infected tissue in the leaf and compare the data using statistics so as to eliminate the bias of observing the symptom subjectively.

8.     In this study we use eye observation in order to compare toxicity effect comparable to control system.

In this study we could not measure toxicity effect by using surface area of infected leaf tissue but the mild, moderate and severe effect were also considered accordingly with total chlorophyll content. 

9.     In the experimental design, is there any positive and negative control of experiment? I cannot draw any conclusion based on the current data without comparing to the control treatment. Besides, the decreased concentration of carbaryl doesn’t mean that such contaminant was absorbed by the plant. The toxicant maybe degraded by the soil microorganisms or by means of photodegradation to become derivatives. Therefore, the author should provide strong evidence to support the fate of carbaryl after experiment (both in root, stem and leave of plants and hydroponic solution) otherwise the data in this current findings is unreliable.

9.     We have control and blank experiment in every experiment.

The control nutrient solution without plant at 0, 5, and 10 mg L-1 of carbaryl concentration was in triplicate and used their concentration for comparison with nutrient solution with plant samples. The carbaryl degradation to 1-naphthol in solution has been added the data in section 4.2 (in the first 4 lines of the paragraph) in the Discussion section.

10.  Back to the basic concept, can carbaryl TOTALLY soluble in water? Can the author tell all the readers about the way to solve the problem relating to the solubility of non-polar organic compound into polar solvent such as water during experiment?

10.  The literature (USDA, 2019) on water solubility of carbaryl has been added in Introduction section.

11.  In figure 4 and 6, why don’t analyze your data in each species separately instead of mixing all data for analysis using ANOVA? Surprisingly, there is NO letter showing significant difference in figure 5a.

11.  In order to compare ability of three different plants (Figure 4-6), we had analyzed statistics by using three plant data sets. The results were comparable and could be concluded on different plant performance. The insignificant difference in Figure 5a might be due to all plants were in stage of growth.

12.  In line 371 to 374, you haven’t show the data of GA, IAA, ABA concentration after experiment in the manuscript. However, the authors tried to explain the experimental results based on GA, IAA, ABA. Therefore, I think author should discuss mainly on the experimental results instead of discussing hypothetical data.

12.  The result data of GA, IAA, and ABA were detected in only a few of plant samples. The experimental results based on GA, IAA, and ABA have been revised in Discussion (Section 4.3) 

The discussion on main experiment results has been added in Discussion (section 4.1)

13.  In figure 7b and figure 7c, is there any trend to show for your experiment? I found there is the same problem as in point 11. Please explain it clearly to the readers about the experimental results related to your current research.

13.  In order to compare ability of three different plants (Figure 7), we had analyzed statistics by using three plant data sets. There were no detectable 1-naphthol in root of all plants, except for sunn hemp root at day 8 cultivation with 10 mg L-1 of carbaryl exposure (0.09 mg kg-1). This result has been added in detail and explained in section 3.3.

14.  In line 458, based on my understanding, there are significant effect on the plant with the root legume colonization in term of nitrogen absorption rather than phosphorus absorption. Please specify the TYPES and SPECIES of legume as well as the status of infection so that it is more specific to absorb phosphate for the plant growth and facilitate phytoextraction.

14.  The review and discussion on the effect on the plant with the root that absorb phosphate and/or nutrient for the plant growth and facilitate phytoextraction has been added in Discussion (the 4th to 12th line of the first paragraph in section 4.3).

15.  The author should delete figure 9 because the content of figure is totally out of the scope for the current research theme.

15.  Figure 9 has been deleted from the manuscript.

Note: Please see supplement data in the file.

Reviewer 3 Report

The seed germination study is inconclusive and detracts from the high overall quality of this manuscript. My recommendation is to delete section 3.5. 

Author Response

Response to Reviewer 3

Phytoextraction potential of sunn hemp, sunflower and marigold for carbaryl contamination: Hydroponic experiment

Response to reviewer’s comment

The manuscript entitled “Phytoextraction potential of sunn hemp, sunflower and marigold for carbaryl contamination: Hydroponic experiment” has been revised according to reviewer’s comment and suggestion. We are appreciated for your kind comment and suggestion. The revised manuscript file has been rechecked English proofread and combined in one file. The edited texts have been in “Track Changes” and our response has been as point by point as follow.

Reviewer’s comment

Authors’ response

The seed germination study is inconclusive and detracts from the high overall quality of this manuscript. My recommendation is to delete section 3.5

The seed germination experiment is important to reveal the survival of the plant after the treatment and toxicity of carbaryl. The discussion on seed experiment has been added.

Round 2

Reviewer 2 Report

After edition, the quality of manuscript substantially improved. There is no additional comments on the manuscript.